# Melatonin as a Repairing Agent in Cadmium- and Free Fatty Acid-Induced Lipotoxicity

**DOI:** 10.3390/biom13121758

**Published:** 2023-12-07

**Authors:** Anna Migni, Francesca Mancuso, Tiziano Baroni, Gabriele Di Sante, Mario Rende, Francesco Galli, Desirée Bartolini

**Affiliations:** 1Department of Pharmaceutical Sciences, University of Perugia, 06123 Perugia, Italy; annamigni4@gmail.com; 2Department of Medicine and Surgery, University of Perugia, 06123 Perugia, Italy; francesca.mancuso@unipg.it (F.M.); tiziano.baroni@unipg.it (T.B.); gabriele.disante@unipg.it (G.D.S.); mario.rende@unipg.it (M.R.)

**Keywords:** melatonin, cadmium, lipotoxicity

## Abstract

(1) Background: Cadmium (Cd) is a potentially toxic element with a long half-life in the human body (20–40 years). Cytotoxicity mechanisms of Cd include increased levels of oxidative stress and apoptotic signaling, and recent studies have suggested that these aspects of Cd toxicity contribute a role in the pathobiology of non-alcoholic fatty liver disease (NAFLD), a highly prevalent ailment associated with hepatic lipotoxicity and an increased generation of reactive oxygen species (ROS). In this study, Cd toxicity and its interplay with fatty acid (FA)-induced lipotoxicity have been studied in intestinal epithelium and liver cells; the cytoprotective function of melatonin (MLT) has been also evaluated. (2) Methods: human liver cells (HepaRG), primary murine hepatocytes and Caco-2 intestinal epithelial cells were exposed to CdCl_2_ before and after induction of lipotoxicity with oleic acid (OA) and/or palmitic acid (PA), and in some experiments, FA was combined with MLT (50 nM) treatment. (3) Results: CdCl_2_ toxicity was associated with ROS induction and reduced cell viability in both the hepatic and intestinal cells. Cd and FA synergized to induce lipid droplet formation and ROS production; the latter was higher for PA compared to OA in liver cells, resulting in a higher reduction in cell viability, especially in HepaRG and primary hepatocytes, whereas CACO-2 cells showed higher resistance to Cd/PA-induced lipotoxicity compared to liver cells. MLT showed significant protection against Cd toxicity either considered alone or combined with FFA-induced lipotoxicity in primary liver cells. (4) Conclusions: Cd and PA combine their pro-oxidant activity to induce lipotoxicity in cellular populations of the gut–liver axis. MLT can be used to lessen the synergistic effect of Cd-PA on cellular ROS formation.

## 1. Introduction

Cadmium (Cd) is a potentially toxic element (PTE) widely distributed in the environment due to anthropogenic activities (industrial processes, mining activities and the combustion of fossil fuels) [1]. Its persistence due to its long half-life and bioaccumulation in living organisms make it a significant public health concern [2]. Human exposure to Cd occurs via ingestion, inhalation and dermal contact [3]. Understanding the sources and exposure routes is crucial for evaluating the potential risks associated with Cd toxicity. Cd exerts its toxic effects through multiple mechanisms. It disrupts cellular homeostasis by interfering with essential biological processes, such as enzymatic activities, antioxidant defense systems and DNA repair mechanisms [4,5,6,7]. Additionally, Cd can induce oxidative stress, leading to the generation of reactive oxygen species (ROS) and subsequent damage to cellular components [8,9]. Cell- and tissue-specific mechanisms of toxicity have been described in different experimental models, with the kidney as a main target of the chronic effects of Cd toxicity and the liver of acute intoxication (reviewed in [10,11,12]).

In the liver cell, Cd exposure reduces cell viability, inducing ROS generation, lipid metabolism abnormalities, lipotoxicity and inflammatory gene activation [13,14,15,16,17,18,19,20,21]. Moreover, fat accumulation in the liver, a condition that can present as either physiological or pathological, increases the tissue-specific accumulation of Cd [22] and may interfere with the stress response and detoxification mechanisms that protect the liver from Cd toxicity (reviewed in [10]). Recent studies in rats demonstrated that Cd modifies the hepatic lipidome inducing de novo lipogenesis and lipoprotein metabolism alterations compatible with the insulin resistance phenotype of metabolic syndrome and non-alcoholic fatty liver disease (NAFLD) [23]. Again, the chronic exposure to Cd in mice fed a high-fat diet accelerates the development of non-alcoholic steatohepatitis (NASH) [21]. NASH is now a main cause of liver transplantation in developed countries and approximately one-fourth of the general population is at risk of developing this form of chronic liver disease [24]. NAFLD pathogenesis depends on a multiple-hit process in which lipotoxicity, an excess of cellular ROS and the activation of inflammatory genes are the earliest instigators of damage and activation of death programs of the liver cells, as well as of other cellular elements, that may contribute to harm the liver health [16,17], including intestinal epithelial cells that represent a natural barrier of the gastrointestinal tract and a system to preserve the liver homeostasis and function. To our knowledge, the effects of lipotoxicity on intestinal epithelial cells remains poorly characterized.

These premises support the hypothesis that Cd may synergize with free fatty acids (FFA) to induce cellular lipotoxicity. The hypothesis was explored in this study using HepaRG human liver cells and CACO-2 intestinal epithelial cells exposed to Cd, and FFA oleic acid (OA) and palmitic acid (PA) in different combinations. Lipotoxicity was investigated, assessing cellular lipid accumulation, cytotoxicity parameters, ROS production and the levels of the inflammatory indicators’ interleukin-6 (IL-6) and LOX-5. Moreover, we tested the study hypothesis using melatonin (MLT), a hormonal regulator of the circadian rhythm, endowed with pharmacological properties as an antioxidant and cytoprotective agent [25,26,27,28,29,30]. Previous studies have demonstrated that MLT provides cytoprotective effects in different models of Cd toxicity [31,32], and preliminary data from our group supported the efficacy of this molecule as a repairing agent in liver cells during combinatorial treatments of Cd and FFA [33], and the pharmacological utilization of MLT was demonstrated to prevent lipotoxicity in experimental models of NAFLD as well as in randomized clinical trials (see [34,35,36] and references therein).

## 2. Materials and Methods

### 2.1. Chemicals

Melatonin (M5250, Sigma-Aldrich, St. Louis, MI, USA), oleic Acid (BioReagent from Sigma-Aldrich, St. Louis, MI, USA; product number: O1383), and palmitic Acid (P0500, Sigma-Aldrich, St. Louis, MI, USA); cadmium chloride (CdCl_2_; 202908, Sigma-Aldrich, St. Louis, MI, USA); PD 98059 (P215; Sigma-Aldrich, St. Louis, MI, USA); 2′,7′-Dichlorofluorescein (35848; Sigma-Aldrich, St. Louis, MI, USA); Oil Red O (O0625; Sigma-Aldrich, St. Louis, MI, USA); MTT (M2003, Sigma-Aldrich, St. Louis, MI, USA); hydrocortisone 21-hemisuccinate (H2882, Supelco, Munich, Germany); insulin (I6634; Sigma-Aldrich); impermeable thiol-reactive AlexaFluor 488 C5 maleimide (AFM, Thermo Fisher Scientific, Waltham, MA, USA).

### 2.2. Cell Cultures and Treatments

#### 2.2.1. Cell Lines and Cell Culture Conditions

HepaRG cells (Thermo Scientific, MA, USA) were immortalized human cells with metabolic and morphological similarities to primary hepatocytes. These cells were maintained in William’s E medium (Sigma-Aldrich) containing 1% GlutaMAX (Invitrogen, Carlsbad, CA, USA), 10% fetal bovine serum (FBS, GIBCO, Life Technologies, Carlsbad, CA, USA), 5 μg/mL insulin (Sigma-Aldrich), and 50 μM hydrocortisone 21-hemisuccinate (Supelco). Experiments were performed between cell passages 18–22.

For 2D differentiation, undifferentiated HepaRG were seeded in gelatin-coated plates and cultured in complete William’s E medium for 48 h until complete fusion. Then, the medium was supplemented with 1.7% (*v*/*v*) DMSO for a further 18 days to allow the undifferentiated HepaRG to mature hepatocytes. We changed the medium every 3 days and captured the optical microscope images to monitor the differentiation stage using a EVOSTM XL Core imaging system for qualitative analysis (Thermo Fisher Scientific).

CACO-2 cells are colon carcinoma cells that grow in monolayers often used to mimic the intestinal epithelial barrier. These were cultured in Dulbecco’s Modified Eagle Medium (DMEM, Thermo Scientific) with the addition of 10% FBS, 1% non-essential ammino acids (NEAA; Euroclone, Milan, Italy) and 1% GlutaMAX (Invitrogen, Carlsbad, CA, USA).

The cell lines were maintained in culture in 96-well plates with a density of 1 × 10^5^ cell/well or in 6-well plates with a density of 2 × 10^5^ cells/m^−2^ in an incubator at 37 °C with a humidified atmosphere of 5% CO_2_.

#### 2.2.2. Primary Hepatocyte Isolation and Culture

Hepatocytes were isolated from 4–6-month-old C57BL/6 J mice purchased from Charles River Laboratories (Calco, Italy). The mice were housed in a pathogen-free facility at the University of Perugia (Perugia, Italy) under controlled light and temperature conditions and treated according to European Community guidelines. The experiments were authorized by the Italian Ministry of Health (protocol number 605/2019-PR). Hepatocyte isolation was performed by two-step perfusion using Liver Perfusion and Liver Digest Media (Life Technologies, Pleasanton, CA, USA) followed by separation with 50% Percoll (GE Healthcare Life Sciences, Pittsburgh, PA, USA) density gradient. Once the liver was washed by perfusion, hepatocytes were dissociated by collagenase, separated from other cells and cultured [37]. The purity of live hepatocytes was routinely monitored by trypan blue exclusion (≥90%). Hepatocytes were cultured in HepatoZYME SFM medium (Gibco). Experimental assays were performed 72 h after hepatocyte plating.

#### 2.2.3. Cell Treatments

In a preliminary series of experiments, the toxicity of Cd was investigated in intestinal and hepatic cells using Experimental Protocol 1 (EP1) of Figure 1. The cells were treated from 4 to 96 h with different concentrations of CdCl_2_ (from 10 to 200 μM). These treatments were performed in the presence or absence of the cytoprotective agent MLT (50 nM). In control experiments of EP1, the cells were treated with vehicles of Cd and/or MLT, namely, 0.001% (*v*/*v*) H_2_O or ethanol (EtOH), respectively. The possible interactive effect between Cd toxicity and FFA-induced lipotoxicity hypothesized in this study was investigated using the EP2 of Figure 1. CACO-2 and liver cells were plated in the 96-well black and transparent color plates (1 × 10^5^ cell/well), and then the cells were pre-treated for 24 h with different concentrations of CdCl_2_ (20 μM and 50 μM for CACO-2 cells; 50 μM and 100 μM for liver cells) and then treated for 48 h with the FFA oleic acid (OA, 200 μM) and/or palmitic acid (PA, 200 μM), or in combination (OA + PA; 200 µM each), in the presence or absence of MLT. OA was dissolved in DMSO and PA was prepared as reported in [38].

For MAPK-ERK1/2 signaling assay, HepaRG and CACO-2 cells were pre-treated for 1 h with ERK inhibitor (PD98059) 50 µM and then treated for 3 h with CdCl_2_ 50 µM or 20 µM, respectively, in the presence or absence of MLT 50 nM.

### 2.3. Cell Viability Assay

Cell toxicity of CdCl_2_ and FFAs and the cytoprotective effect of MLT were determined by MTT test (Sigma-Aldrich) in 96-well plates with a seeding density of 10,000 cells/cm^2^, as reported in [39]. Briefly, after different times of incubation (3 to 72 h), the cell medium was replaced with fresh medium containing the MTT solution (1:10 *v*/*v*) and the cells were incubated for 2.5 h at 37 °C. An MTT solubilization solution (0.1 N HCl containing 10% vol/vol Triton X-100 in anhydrous isopropanol) was used to dissolve the formazan crystals formed during the incubation by the activity of mitochondrial dehydrogenases of viable cells, and a microplate reader (DTX880 Multimode Detector, Beckman Coulter, Brea, CA, USA) was used to measure the absorbance of the cell culture medium at 570 nm. The results were expressed as percentage of the optical density (OD) observed in control cells.

### 2.4. Apoptosis Assay by Flow Cytometry

Each group of cells was treated with Trypsin-EDTA solution (Sigma-Aldrich) to create a single-cell suspension, centrifuged (1500 rpm, 5 min) and washed twice with PBS solution, resuspended with 500 μL of binding buffer, then mixed sequentially with 5 μL of FITC-labeled Annexin V and 5 μL of PI stain, and incubated for 10 min at room temperature, and then immediately, cell apoptosis was measured using BD Accuri C6 Plus System (BD Biosciences, San Jose, CA, USA). An Annexin V-FITC/PI Apoptosis Kit was purchased from Elabscience (Cat.No.: E-CK-A211).

### 2.5. Intracellular and Extracellular ROS Analysis

Intracellular ROS were measured by the oxidative conversion of the intracellular probe 2′,7′-dichlorofluorescein-diacetate (DCFH-DA; Sigma-Aldrich) to the fluorescent derivative 2′,7′-dichlorofluorescein (DCF). Briefly, the cells were treated with 50 µM dichloro-fluorescein diacetate (DCFH-DA) in PBS and incubated in the dark for 30 min at 37 °C to allow the probe to enter the cell and react with cellular ROS after enzymatic hydrolysis by intracellular esterases to its derivative dichloro-fluorescein (DCFH). This is a non-fluorescent derivative that, other than losing the ability to cross the membrane, is reactive to cellular ROS to produce a highly fluorescent form of the probe, i.e., the oxidized DCFH derivative. After washing with PBS to remove the non-internalized DCFH-DA, the fluorescence of the cells was recorded using a microplate reader set with excitation λ of 485 nm and emission λ of 535 nm. The obtained fluorescence (FL) was normalized against the absorbance (Abs) of the MTT test (FL/Abs), and the results were expressed as the percentage of the control cell value (identified as CTL).

Extracellular H_2_O_2_ was determined using a colorimetric assay kit (Elabscience, Houston, TX, USA) adapted to a 96-well microplate reader. A total of 100 μL of Reagent 1 (buffer solution of Elabscience kit) was pipetted in each well and incubated at 37 °C for 10 min. Then, 10 μL of the cell supernatant or of the negative and positive control test (namely, bi-distilled water and 60 mmol/L H_2_O_2_, respectively) were added, followed by 100 μL of Reagent 2 (ammonium molybdate); the absorbance of the solution was recorded at 405 nm as optical density units (OD) in a DTX880 Multimode Detector microplate reader (Beckman Coulter). Hydrogen peroxide concentrations were calculated using the following formula: [H_2_O_2_] (mmol/L) = (ΔA1/ΔA2) × c × f.
ΔA1: ODSample—ODBlank;ΔA2: ODStandard—ODBlank;c: H_2_O_2_ concentration of the standard sample = 60 mmol/L;f: dilution factor of sample.

### 2.6. Oil Red O (ORO) Assay

Cellular lipids were measured by Oil Red O (ORO) staining according to the procedure described in [38]. Briefly, CACO-2 and HepaRG cells were fixed with 10% neutral buffer formalin (Leica) for 30 min; the fixed cells were thus washed twice with sterile bi-distilled water and then incubated with 60% isopropanol for 5 min. The cells were stained with ORO solution for 2–5 min and then washed four times with sterile bi-distilled water before staining with hematoxylin solution (Sigma-Aldrich, St. Louis, MI, USA) for 1 min. The stained cells were washed again with sterile bi-distilled water and then were assessed by optical microscopy using a EVOSTM XL Core imaging system for qualitative analysis (Thermo Fisher). Furthermore, to quantify the cellular content of ORO, the cell pellet was incubated for 10 min with 100 μL of isopropanol, and the absorbance of the extract was assessed at 510 nm using a multiplate reader monochromator (TECAN, Männedorf, Switzerland).

### 2.7. Analysis of Cell-Surface Thiols by Flow Cytometry

After treatments, HepaRG cells were recovered and washed with PBS twice, then incubated with 10 µM AlexaFluor 488 C5 maleimide (AFM) probe in PBS for 30 min at 37 °C. After having been washed again in PBS, the cells were analyzed by flow cytometry using BD Accuri C6 Plus System (BD Biosciences, San Jose, CA, USA). Dead cells were gated out by staining with Propidium Iodide (PI, Thermo Fisher Scientific, Waltham, MA, USA), and 1 × 10^4^ living cells was analyzed. The results were expressed as mean fluorescence of the AFM probe.

### 2.8. Total Protein Extraction and Quantification

The cells were plated in 6-well plates, and proteins were extracted with 100 μL of cell lysis buffer (Cell Signaling Technology Inc., Danvers, MA, USA) and protease inhibitor cocktail (Pierce, Thermo Fisher Scientific Inc., Waltham, MA, USA). The cell lysates were maintained in ice for 40 min before centrifugation at 14,000 rpm for 20 min at 4 °C, and the supernatants containing the total protein extract were collected and stored at –80 °C before determination. Protein concentrations were measured by bicinchoninic acid assay (BCA assay, Pierce, Thermo Fisher Scientific Inc., Waltham, MA, USA). A total of 200 μL of a solution 1:50 (*v*/*v*) of reagent A (bicinchoninc acid, sodium bicarbonate, sodium tartrate and sodium carbonate in 0.1 N NaOH, 11.25 final pH) and reagent B (4% *w*/*v* of CuSO_4_·5 H_2_O in water) were pipetted in a 96-well plate together with 10 μL of a sample or bovine serum albumin (BSA) at different concentration that was used as external standard for assay calibration. After 30 min of incubation at 37 °C in the dark, the absorbance of the samples and analytical standard was recorded at 570 nm using a microplate reader (DTX880 Multimode Detector, Beckman Coulter, Brea, CA, USA).

### 2.9. SDS-PAGE and Immunoblotting

Cell proteins (10–30 μg) were separated by 10–12% sodium dodecyl sulfate–polyacrylamide gel electrophoresis (SDS–PAGE) and then transferred to a nitrocellulose membrane (Millipore, Billerica, MA, USA) for immunoblot analysis. Membranes were blocked for 2 h at room temperature with 5% skim milk (Sigma-Aldrich) in Tris-buffered saline and 0.1% Tween20 (TBST) and then incubated overnight at 4 °C with the primary antibodies (Cell Signaling Technology, Danvers, MA, USA) that included the following: phospho-p44/42 MAPK (ERK1/2) rabbit monoclonal antibody (mAb) (1:1000), pp42/44 MAPK (Erk1/2) rabbit mAb (1:1000), phospho-SAPK/JNK rabbit mAb (1:1000), SAPK/JNK rabbit mAb (1:1000), GAPDH rabbit mAb (1:1000), phospho-p38 rabbit mAb (1:1000), p38 rabbit mAb (1:1000), Catalase rabbit mAb (1:1000), SOD-2 rabbit mAb (1:1000), 5-LOX rabbit mAb (1:1000) and α-Tubulin rabbit mAb (1:1000). The day after the incubation, the membranes were washed 3 times with TBST 0.1% and incubated with anti-rabbit or anti-mouse IgG (1:2000) horseradish peroxidase-linked secondary antibodies (Cell Signaling Technology Inc., Danvers, MA, USA). Protein bands were detected using an ECL Clarity Max (BioRad, Hercules, CA, USA), and band quantification was performed with a Gel-Pro Analyzer.

### 2.10. IL-6 Analysis

IL-6 levels in cell culture media of liver cells were detected using a commercial IL-6 ELISA kit (cat. no. BMS603-2) from eBioscience (Thermo Fisher Scientific, Inc., MA, USA) following the manufacturer’s instructions.

### 2.11. Statistical Analyses

Statistical comparisons were performed using one-way ANOVA test. Data were expressed as mean ± SD of 3 independent experiments. The probability of error accepted for significant differences was *p* < 0.05 (* or #), and highly significant differences were identified for *p* < 0.01 (** or ##). Data analysis and graphical presentation were performed with GraphPad Prism 9 (Version 9.0.2).

## 3. Results

### 3.1. CdCl_2_ Toxicity and Cytoprotective Effect of MLT

CdCl_2_ toxicity in the liver and intestinal cell models utilized in this study, namely, primary murine hepatocytes, HepaRG cells before and after differentiation to hepatocyte-like cells, and CACO-2 intestinal epithelial cells, were investigated using the treatment protocol EP1 (Figure 1). MTT test data showed a concentration- and time-dependent toxicity of CdCl_2_ in all of the cell models (Appendix A), with a significant reduction in cell viability levels after 24 h of treatment with Cd concentrations > 20 μM. Cytofluorimetry data demonstrated that this toxicity effect of Cd, assessed at 48 h, is associated with a concentration-dependent induction of apoptosis and necrotic cell death in both the HepaRG and CACO-2 cell lines (Figure 2A and Figure 2B, respectively). However, Cd toxicity in CACO-2 cells was characterized by higher levels of late apoptosis and necrosis compared to HepaRG cells (Figure 2). These cell death data indicate that CACO-2 cells are more prone than HepaRG cells to Cd toxicity.

MLT produced a significant cytoprotective effect, improving cell viability data of both the differentiated and undifferentiated form of HepaRG cells (Appendix A and Appendix A, respectively), as well as in CACO-2 cells (Appendix A) exposed to CdCl_2_ toxicity. Cell viability was also improved in primary murine hepatocytes, in which MLT exploited a protective effect both during co-treatment (Appendix A) and after treatment with Cd (Appendix A). Apoptosis and necrosis data confirmed that this cytoprotective activity of MLT depends on a significant reduction in cell death levels in both the two cell lines (Figure 2).

The expression and activity of the survival protein kinase MAPK-ERK1/2 and the stress-activated protein kinases (SAPKs) p38 and JNK were investigated in HepaRG cells exposed to Cd toxicity, since these control cell survival and death pathways during the exposure to cellular stressors also mediating the MLT cytoprotective function [40]. Cd toxicity in HepaRG cells was associated with a significant increase in phosphorylation levels of all protein kinases, namely, MAPK-ERK1/2 (Figure 3A), JNK (Figure 3B) and p38 (Figure 3C), indicating the role of these kinases in the Cd toxicity and its effects on the balance between survival and death pathways of the liver cell.

MLT was confirmed to behave as a potent ERK1/2 agonist [41] by the stimulation of both the protein expression and phosphorylation of MAPK in HepaRG cells (Figure 3A), which is a specific effect as demonstrated by the utilization of the pharmacological inhibitor PD98059 (Appendix A). However, during Cd exposure, MLT counteracted MAPK-ERK1/2 and SAPKs (p38 and JNK) activation of this liver cell line (Figure 3A–C), also restoring stress response parameters affected by the activity of these kinases, including the redox indicators cellular surface thiols and ROS (Figure 3D and Figure 3E, respectively), the antioxidant enzymes catalase (CAT) and superoxide dismutase 2 (SOD2) (Appendix A), and the inflammatory enzyme 5-LOX (Appendix A). The MLT activation effect on the antioxidant enzymes CAT and SOD2 was also confirmed in intestinal cells (Appendix A).

### 3.2. Effect of CdCl_2_ and FFA Treatments on Cellular Lipid Accumulation

The effects of Cd and FFA on lipid levels of the different cell models were studied using the lipid probe ORO in EP2 of Figure 1.

In undifferentiated HepaRG cells, both Cd and FFA, when investigated as separate and independent treatments, were found to induce lipid accumulation in droplets (see light microscopy images of Figure 4A and spectrophotometric data of Figure 4B). This lipid accumulation was associated with a marked reduction in the cell mass revealed by light microscopy images (Figure 4A), demonstrating induction of lipotoxicity by these treatments that was concentration-dependent with respect to CdCl_2_ in this human liver cell line. When CdCl_2_ and FFA treatments were combined, these synergized to exacerbate their lipid accumulation and cell mass reduction effects with PA, which was confirmed to induce a stronger interaction with CdCl_2_ compared to OA. Rather, the latter FFA species appeared to mitigate the lipid accumulation effect of PA in the OA + PA treatment (Figure 4B) but not its lipotoxicity effect, as assessed by the number of cells remaining in the light microscopy fields of Figure 4A.

The concentration-dependent effect of Cd on neutral lipid accumulation was also confirmed in primary mouse hepatocytes studied by spectrophotometric analysis of cellular ORO (Figure 5).

Light microscopy images and spectrophotometric data of CACO-2 cells (Figure 6) demonstrate that this cell line is less susceptible to the lipid accumulation effects of Cd and FFA compared to liver cells (Figure 4 and Figure 5).

### 3.3. Cell Death Levels in Cd- and FFA-Induced Lipotoxicity, and MLT Cytoprotective Effect

Cell viability (Appendix A) and cell death data (Figure 7) demonstrated the higher toxicity of PA compared to OA in both the HepaRG and CACO-2 cell lines; rather, OA appears to protect these cells form PA toxicity in combinatorial treatments (i.e., OA + PA). When combined with Cd exposure in the EP2 of Figure 1, FFA increased their toxicity with a higher reduction in viability and increased cell death levels in both the two cell lines, suggesting synergizing effects of Cd and FFA in inducing lipotoxicity; cell viability data confirmed this synergy of effects also in primary murine hepatocytes (Figure 5E).

MLT lessened the cell death induction effect of the different combinations of Cd and FFA in HepaRG and CACO-2 cells (Figure 7A–D), whereas the cell viability reduction effect was not significantly affected (Appendix A), indicating different sensitivity of these cell toxicity assays. However, MLT was found to protect the mouse primary hepatocytes from the cell viability reduction effect of Cd exposure combined with OA + PA treatment (Figure 5E), and this effect of MLT was in good agreement with that of lipid accumulation of these liver cells (Figure 5D).

### 3.4. Intracellular and Extracellular ROS

An increased production of cellular ROS is considered a causal indicator of the cytotoxicity process of both the Cd exposure and FFA-induced lipotoxicity. When ROS were studied in HepaRG cells (Figure 8A), the exposure to Cd or FFA increased their levels, with PA generating the most potent ROS generating response; important enough was the observation that Cd synergized with FFA to induce this oxygen activation effect in these cells, which is consistent with other indicators of lipotoxicity shown earlier in Section 3.2 and Section 3.3. FFA synergized their ROS production effect with Cd also in CACO-2 intestinal cells (Figure 8B), and MLT was very effective in reducing this combined effect of Cd and FFA treatment in both of the two cell lines (Figure 8A and Figure 8B, respectively).

In primary murine hepatocytes, Cd partially synergized with OA + PA to induce cellular oxygen activation (Figure 8C) and H_2_O_2_ efflux (Figure 8D), and MLT significantly reduced these effects.

### 3.5. Effect of CdCl_2_ and MLT on IL-6 Secretion Levels of HepaRG Cells and Primary Murine Hepatocytes

A dose-dependent induction effect of CdCl_2_ was observed for IL-6 secretion in both HepaRG cells and primary murine hepatocytes, which was further enhanced by the combination of OA + PA treatment (Figure 9). MLT significantly reduced the effect of Cd and FFA on the levels of this pro-inflammatory cytokine in murine primary hepatocytes. 

## 4. Discussion

Cellular toxicity mechanisms of Cd include the induction of mitochondrial damage and increased ROS production, defects of antioxidant defenses, activation of inflammatory genes and cell death programs [4,42]. Another mechanism recently identified for this PTE is the induction of cellular lipotoxicity, a specific process of lipid accumulation deriving from an increased cellular flux of FFA and activation of lipid biosynthesis and redistribution processes. This results in a lipid excess stored in lipid droplets, a functionally specialized vesicular system of lysosomal origin that characterizes the “fatty” phenotype of some tissues, especially the liver [40]. These cellular effects of Cd exposure are associated with the induction of lipid metabolism alterations and oxidative stress, which have been documented in in vitro studies on human and animal liver cells [13,14,15,16,17,18,19,20,21] as well as in vivo in animal models of chronic exposure to Cd [21,23]. Alterations of the cellular lipidome recently characterized in mouse liver and human hepatocarcinoma cells demonstrate that Cd toxicity induces specific changes in membrane phospholipids, especially in phosphatidylcholine synthesis and remodeling, also increasing the relative abundance of arachidonic acid residues in complex lipids [43], which is a characteristic lipidomic hallmark of inflammation and a potential therapeutic target of human fatty liver disease [44].

These pieces of evidence stimulated us to explore the hypothesis that Cd may synergize with other factors to induce lipotoxicity; these include FFA that have a major pathogenic role in NAFLD (reviewed in [40,44]). In this in vitro study, we utilized OA and PA since their supplementation to human liver cells, performed at different concentrations and times of exposure [38,45], has consistently been demonstrated to induce lipotoxicity and all of the other hepatocellular, molecular and metabolic features of NAFLD. This experimental model has been utilized to explore both lipotoxicity mechanisms and the efficacy of pharmacological therapies and nutraceuticals at the pre-clinical level [46,47]. We also studied Cd and FFA toxicity in CACO-2 intestinal epithelial cells since, to our knowledge, the effects of lipotoxicity on the gut epithelium remain poorly characterized, although potentially relevant for the liver health [40].

Our results confirm that Cd and FFA synergize their lipotoxicity effects in liver cells, including mouse primary hepatocytes and the human pre-hepatocyte cell line HepaRG that was studied either before or after differentiation to mature liver cells. These effects were confirmed for all of the lipotoxicity hallmarks investigated in this study (shown from Figure 4 onward), including lipid droplet formation, ROS production, antioxidant enzyme expression, inflammatory parameters such as 5-LOX expression and IL-6 secretion, a reduction in cell viability, induction of cell death by apoptosis and necrosis, and a loss of cellular elements as assessed through direct microscopy observation.

CACO-2 intestinal epithelial cells also develop cues of lipotoxicity when exposed to Cd or to a combination of Cd and FFA, even if the levels of cellular lipid accumulation were lower compared to those observed in the liver cells (Figure 5).

Furthermore, we utilized MLT to explore whether the synergistic effects of Cd and FFA as lipotoxicity inducers can be prevented in the liver cell. This molecule was selected because it is a cytoprotective agent with proven efficacy in lessening Cd toxicity effects in different experimental models ([31,32] and references therein); moreover, it is effective in preventing liver damage in experimental models of fatty liver disease as well as in clinical trials on NAFLD patients ([34,35,36] and references therein).

As an important finding in the present study, MLT was found to protect both HepaRG and primary hepatocytes from the pro-oxidant and pro-inflammatory (Figure 7, Figure 9 and Appendix A) effect of both individual and combined treatments with Cd and FFA, significantly reducing cytotoxicity in HepaRG and CACO-2 cells (Figure 6 and Appendix A) and primary murine hepatocyte (Figure 5E). These results confirm the homeostatic properties of this molecule in Cd toxicity, highlighting its pleiotropic activity as a cytoprotective agent in cellular models of Cd-induced lipotoxicity, which is a main finding of this study. Herein, we demonstrate that MLT controls the effects of Cd exposure via MAPKs modulation (Figure 3). Cd is known to activate ERK and STAT3, and these are key players of the cellular stress responses [10,48,49,50] that can be modulated by MLT. This hormonal substance is a potent MAPK-ERK1/2 agonist, with a key role in cell cycle regulation and death signaling during the stress response to ROS and inflammatory factors [41]. In this respect, it is worth noting the inhibitory effect of a pharmacological dose of MLT (50 nM) on the induction response to Cd exposure of SAPKs and MAPK-ERK of liver cells (Figure 3 and Appendix A); this provides a mechanistic explanation to the antioxidant and anti-inflammatory activity of MLT, as well as to its cytoprotective effect on liver cells exposed to Cd and FFA-induced lipotoxicity.

The use of “supraphysiological” concentrations of Cd and melatonin could be interpreted as a limit of this in vitro study; however, these concentrations are commonly utilized in vitro to obtain significant responses in liver cells and their relevance is discussed considering in vivo data. In accordance with this, MLT was found to improve hepatometabolic indices of NAFLD in obese mice fed a high-fat diet via MAPK-JNK/P38 signaling modulation [36], and reduced levels of pro-inflammatory cytokines (as TNFα, Il-6 and Il-1β) and an inducible form of the NO-generating enzyme iNos in the liver of mice exposed to Cd toxicity [51], which appears to support our mechanistic interpretation.

## 5. Conclusions

Here, we demonstrate that Cd can synergize with FFA to induce lipotoxicity effects in the liver and intestinal epithelial, which is a potentially relevant pathogenic mechanism of fatty liver disease [45]. Mechanistic aspects of such a toxicological interaction include the stimulation of pro-oxidant and inflammatory pathways that can be mitigated by MLT. The cytoprotective properties of this molecule are worth investigating at the pre-clinical and clinical level for an application protocol of hepatoprotection and therapy of hepatic lipotoxicity in Cd exposure.

## Figures and Tables

**Figure 1 biomolecules-13-01758-f001:**
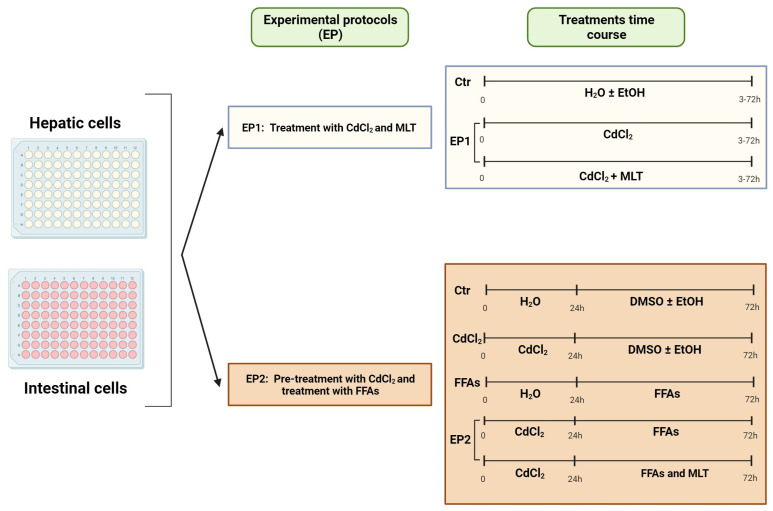
Experimental protocols (EP). The scheme shows the two-arm EP used in this study; namely, EP1 was designed to assess CdCl_2_ cytotoxicity effects as studied by the cell viability test and the activity of survival and stress MAPKs. EP2 was set to explore the hypothesis that Cd can interact with FFA in inducing lipotoxicity and ROS generation in liver and intestinal cells. In EP2, the cells were pre-treated with different concentrations of CdCl_2_ for 24 h and then treated for 48 h with 200 μM final concentration of the FFA oleic acid (OA) or palmitic acid (PA) or a combination of both (OA + PA). Effects of Cd and FFA were studied either in the presence or absence of the cytoprotective molecule MLT (50 nM) and against control experiments run with the vehicles of these treatments. Further details on experimental conditions are reported in the text.

**Figure 2 biomolecules-13-01758-f002:**
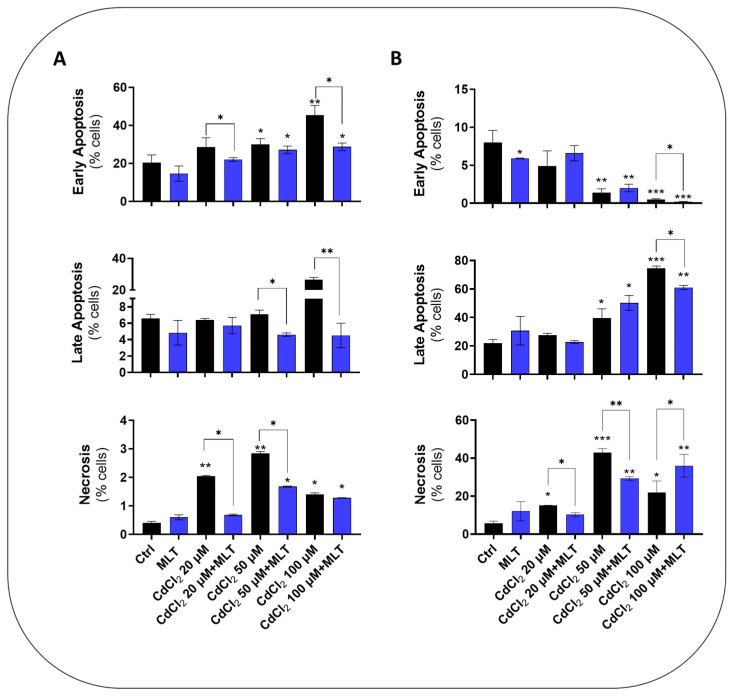
Cell death levels in HepaRG (**A**) and CACO-2 (**B**) cells treated with CdCl_2_ and MLT. Apoptosis assay was studied by flow cytometry analysis after 48 h of treatment with different concentrations of CdCl_2_ (as shown in the chart) and 50 nM MLT using EP1 of Figure 1. Apoptosis and necrosis levels were determined by staining the cells by annexin V-FITC and propidium iodide (PI). Data were mean ± SD of three independent experiments. One-way ANOVA test: * *p* < 0,05; ** *p* < 0.001; *** *p* < 0.0001 (control vs. all treatments) and (CdCl_2_ vs. CdCl_2_ + MLT).

**Figure 3 biomolecules-13-01758-f003:**
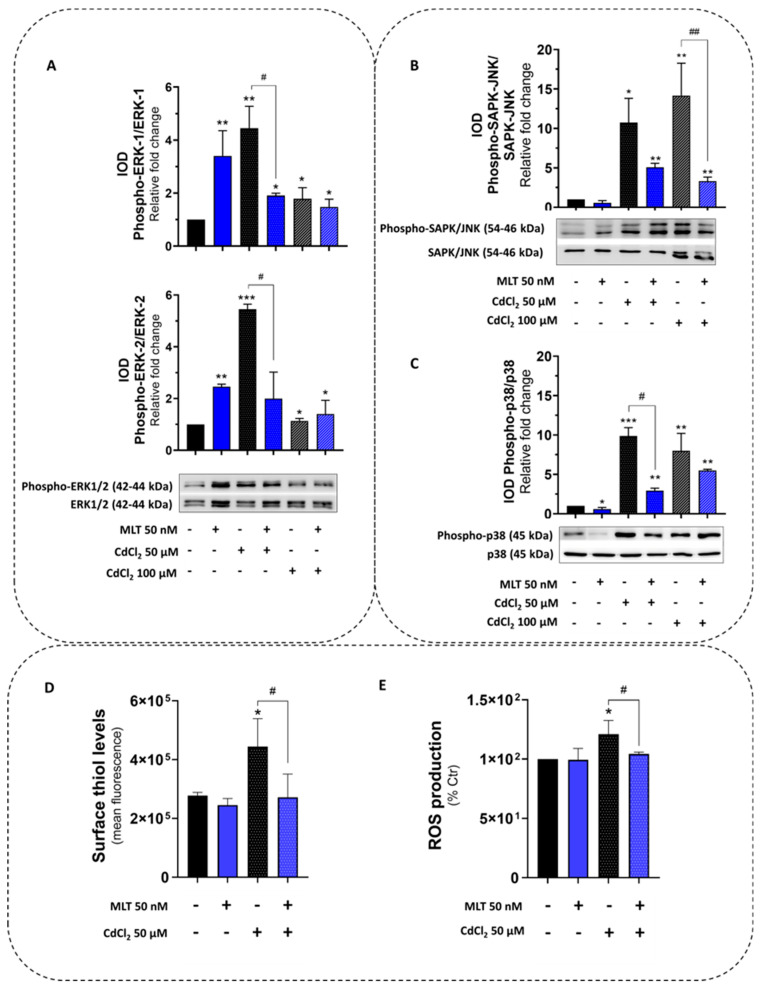
Protein kinase activity and redox parameters of HepaRG cells treated with CdCl_2_ and MLT. (**A**) MAPK-ERK1-2, (**B**) SAPK/JNK and (**C**) p38-MAPK activity was studied by immunoblot assessing phosphorylation and the native form of the proteins. (**D**) Cell-surface thiols and (**E**) cellular ROS were assessed by FACS-scan analysis. HepaRG cells were exposed for 3 h to CdCl_2_ (50 and 100 µM) and MLT (50 nM) that were studied as both separate treatments and in co-treatment mode (EP1). One-way ANOVA test: * *p* < 0.05; ** *p* < 0.01; *** *p* < 0.001 (control vs. all treatments). # *p* < 0.05; ## *p* < 0.01 (CdCl_2_ vs. CdCl_2_ + MLT).

**Figure 4 biomolecules-13-01758-f004:**
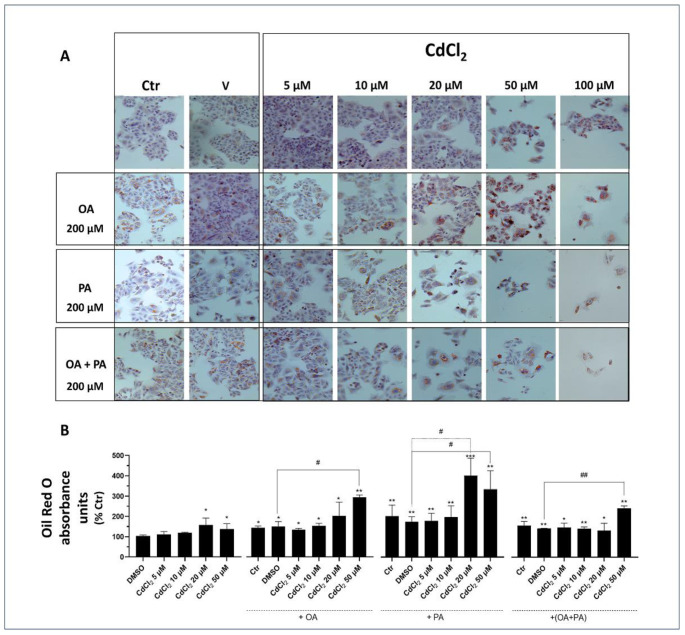
Effect of CdCl_2_ and FFAs on cellular lipid accumulation in undifferentiated human liver HepaRG cells. Cell treatments were performed with the Experimental Protocol EP2 described in detail in Figure 1 and in the section “Methods”. Briefly, the cells were studied after a 24 h pre-treatment with CdCl_2_ and 48 h treatment with FFAs (200 µM final concentration each). (**A**) Hematoxylin and Oil Red O (ORO) were used to stain liver cells and lipid droplets, respectively (40× magnification). (**B**) Quantification of cellular lipids by spectrophotometric determination of ORO absorbance at 510 nm. The FFAs used for the treatments were oleic acid (OA), palmitic acid (PA) and their combination (OA + PA). One-way ANOVA test: * *p* < 0.05; ** *p* < 0.01; *** *p* < 0.001 (control test vs. all treatments); # *p* < 0.05; ## *p* < 0.01 (FFAs vs. all treatments).

**Figure 5 biomolecules-13-01758-f005:**
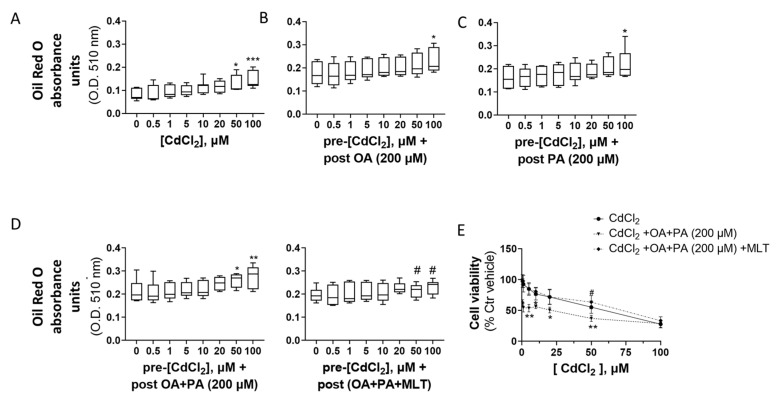
Effect CdCl_2_, FFA and MLT on cellular lipids (**A**–**D**) and cell viability levels of primary murine hepatocytes. Cellular lipids were assessed by ORO staining and spectrophotometric analysis at 510 nm of cellular extracts, as described in detail in the section “Materials and Methods”. (**A**) Lipids were assessed after 24 h pre-treatment with CdCl_2_ and 48 h treatment with different FFAs (200 µM final concentration each), namely, OA (**B**), PA (**C**) and OA + PA tested alone or in combination with the cytoprotective agent MLT (50 nM) (**D**). One-way ANOVA test: * *p* < 0.05; ** *p* < 0.01, *** *p* < 0.001 (control test vs. all treatments); # *p* < 0.05 (FFAs vs. FFAs/MLT). Cell viability was measured by MTT assay as described in detail in the section “Materials and Methods” (**E**). The cells were pre-treated for 24 h with increasing concentrations of CdCl_2_ and then treated with FFAs (200 µM each) for 48 h in the presence or absence of the cytoprotective agent MLT (50 nm) (**E**). One-way ANOVA test: * *p* < 0.05; ** *p* < 0.01 (CdCl_2_ + OA + PA vs. all treatments); # *p* < 0.05 (CdCl_2_ + OA + PA vs. CdCl_2_ + OA + PA + MLT).

**Figure 6 biomolecules-13-01758-f006:**
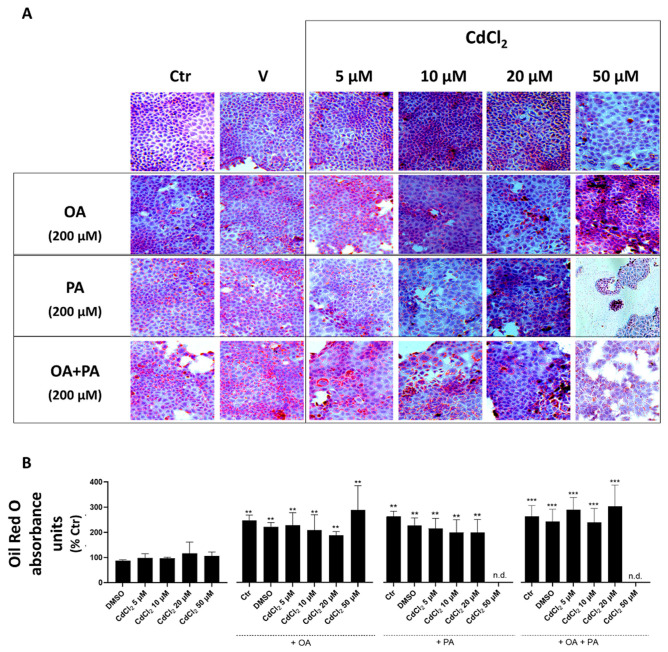
Effect of CdCl_2_ and FFAs on the accumulation of cellular lipids in CACO-2 intestinal epithelial cells. Cell treatments were as described in the legend of Figure 7 (Experimental Protocol EP2, Figure 1). (**A**) Hematoxylin and Oil Red O (ORO) were used to stain liver cells and lipid droplets, respectively (40× magnification). (**B**) Quantification of cellular lipids by spectrophotometric determination of ORO absorbance at 510 nm. The FFAs used for the treatments were oleic acid (OA), palmitic acid (PA) and a combination of the two (OA + PA). One-way ANOVA test: ** *p* < 0.01; *** *p* < 0.001 (control test vs. all treatments). n.d. (non-detectable).

**Figure 7 biomolecules-13-01758-f007:**
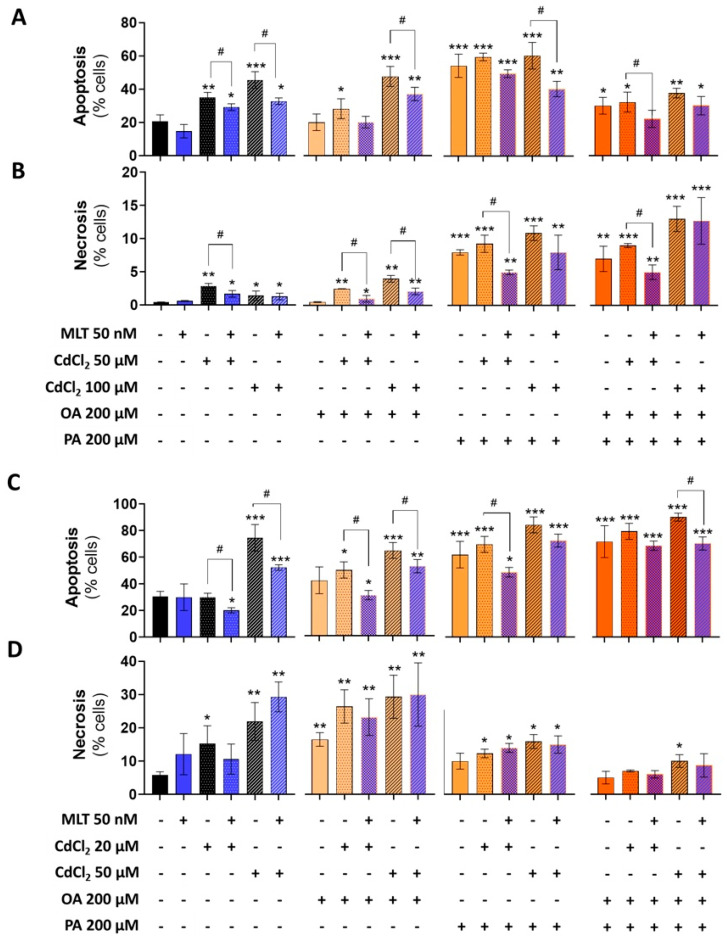
Effect of CdCl_2_, FFA and MLT on cell death levels of human liver HepaRG cells and CACO-2 intestinal epithelial cells. Apoptosis and necrosis levels were studied in HepaRG (**A**,**B**) and CACO-2 (**C**,**D**) cells exposed to different treatments, as described in EP2 (Figure 1). Apoptosis and necrosis were measured by FACS-scan using Annexin-V and Propidium Iodide probes as described in the methods section. One-way ANOVA test: * *p* < 0.05; ** *p* < 0.01; *** *p* < 0.001 (control vs. all treatments). # *p* < 0.05; (CdCl_2_ vs. all treatments); # *p* < 0.05 (CdCl_2_ and/or FFAs vs. CdCl_2_ and/or FFAs + MLT).

**Figure 8 biomolecules-13-01758-f008:**
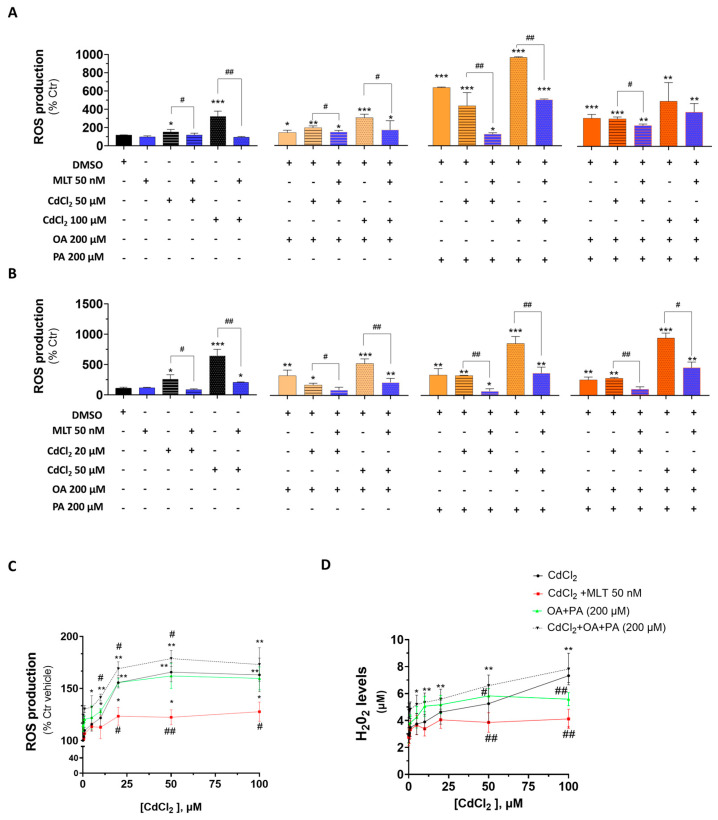
Cellular ROS levels in liver and intestinal cells treated with CdCl_2_, FFAs and the cytoprotective agent MLT. Cellular ROS were measured in HepaRG (**A**), CACO-2 (**B**) and primary murine hepatocytes (**C**); extracellular levels of H_2_O_2_ (**D**) were also determined in primary murine hepatocytes only. Cd, FFA and MLT treatments of HepaRG and CACO-2 cells were performed according to EP2 (Figure 1), whereas primary murine hepatocytes were pre-treated for 24 h with increasing concentrations of CdCl_2_ and then treated for 48 h with FFAs (200 µM final concentration each) in the presence or absence of the cytoprotective agent MLT (50 nm). One-way ANOVA test: (**A**,**B**) * *p* < 0.05; ** *p* < 0.01; *** *p* < 0.001 (control vs. all treatments). # *p* < 0.05 (CdCl_2_ vs. all treatments); # *p* < 0.05; ## *p* < 0.01 (CdCl_2_ and/or FFAs vs. CdCl_2_ and/or FFAs +MLT); (**C**,**D**) # *p* < 0.05; ## *p* < 0.01 (CdCl_2_ vs. CdCl_2_ + MLT and OA + PA vs. CdCl_2_ + OA + PA).

**Figure 9 biomolecules-13-01758-f009:**
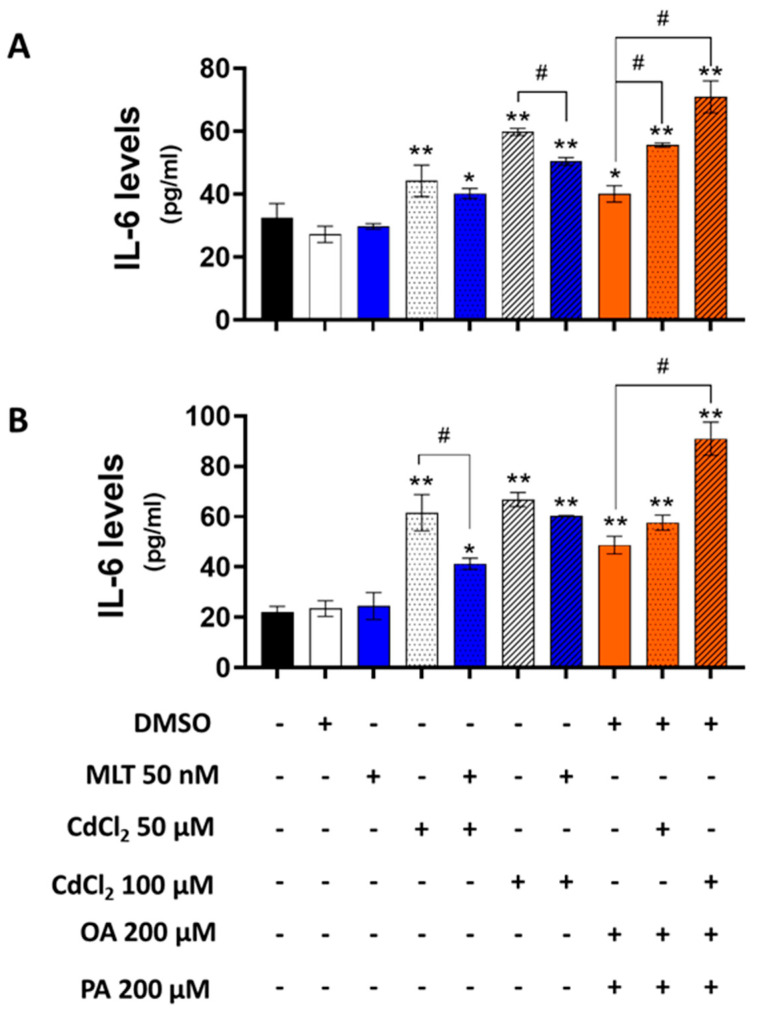
Levels of IL-6 in the culture medium of HepaRG cells (**A**) and primary murine hepatocytes (**B**) treated with CdCl_2_, FFAs and MLT. The experimental protocol for cell treatments was EP2 of Figure 1. IL-6 (pg/mL) levels were measured by ELISA method in cell culture media. Data were mean ± SD of 3 independent experiments. One way ANOVA test: * *p* < 0.05, ** *p* < 0.01 control (or DMSO) versus all treatments; # *p* < 0.05 CdCl_2_ versus CdCl_2_ + MLT or OA + PA versus CdCl_2_ + OA + PA.

## Data Availability

The data presented in this study are available on request from the corresponding author.

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
