# Peer review of "Melatonin as a Repairing Agent in Cadmium- and Free Fatty Acid-Induced Lipotoxicity"

_biomolecules, 2023, doi:10.3390/biom13121758_

Round 1

Reviewer 1 Report

Comments and Suggestions for Authors

The paper investigates the cytotoxic effects of cadmium (Cd), a potentially toxic element, on intestinal epithelium and liver cells, with a focus on its interaction with fatty acids (FA)-induced lipotoxicity. Cd exposure is associated with oxidative stress and reduced cell viability in both cell types. Cd and FA combined to induce lipid droplet formation and reactive oxygen species (ROS) production, with palmitic acid (PA) having a stronger effect than oleic acid (OA) in liver cells. Melatonin (MLT) exhibited cytoprotective properties against Cd toxicity in primary liver cells.

1. Why is cadmium (Cd) considered a significant public health concern, and what are its common sources of exposure?

2. What are the cytotoxic effects of Cd on liver and intestinal cells, and how do these effects relate to non-alcoholic fatty liver disease (NAFLD)?

3. How does the combination of Cd and fatty acids (FA) affect lipid droplet formation and ROS production in liver cells?

4. What role does melatonin (MLT) play in protecting cells from Cd toxicity, and how does it interact with Cd-induced oxidative stress?

5. How do the results of this study contribute to our understanding of Cd toxicity and its impact on the gut-liver axis in the context of lipotoxicity and oxidative stress?

Author Response

The paper investigates the cytotoxic effects of cadmium (Cd), a potentially toxic element, on intestinal epithelium and liver cells, with a focus on its interaction with fatty acids (FA)-induced lipotoxicity. Cd exposure is associated with oxidative stress and reduced cell viability in both cell types. Cd and FA combined to induce lipid droplet formation and reactive oxygen species (ROS) production, with palmitic acid (PA) having a stronger effect than oleic acid (OA) in liver cells. Melatonin (MLT) exhibited cytoprotective properties against Cd toxicity in primary liver cells.

  1. Why is cadmium (Cd) considered a significant public health concern, and what are its common sources of exposure?

Answer 1:

We acknowledge the Reviewer for the interest on our study and the specific query.

Health concerns and sources of exposure are described in the first paragraph of the Introduction. Further information on these aspects is reported below for the Reviewer consultation and in case of specific interests for this topic.

Cadmium (Cd) is considered a significant Public Health concern due to its toxicity and the potential for long-term health effects when people are exposed to it. Cd in fact is a heavy metal that is toxic to humans. It can accumulate in the body over time, leading to adverse health effects. Moreover, long-term exposure to cadmium has been linked to an increased risk of lung cancer, especially among smokers. Cd is present in tobacco smoke, making smoking a major source of exposure. Cd is known to target the kidneys. Prolonged exposure can lead to kidney damage and may result in a condition known as cadmium-induced nephropathy. Cd can also accumulate in bones, which can lead to a weakening of the skeletal system. This is particularly concerning for individuals with high levels of exposure. Some studies have suggested that Cd exposure may be associated with cardiovascular problems, including hypertension and an increased risk of heart disease.

Common sources of Cd exposure include:

- Tobacco smoke: Smoking or exposure to second-hand smoke is a significant source of Cd exposure.

- Diet: Cd can be found in some foods, especially in shellfish, grains, and leafy vegetables. It can accumulate in plants from contaminated soil.

- Occupational exposure: Laborers in industries such as battery manufacturing, mining, and metal plating may be exposed to high levels of Cd if proper safety measures are not in place.

- Contaminated water: In some cases, Cd may leach into water sources from natural deposits or industrial pollution, leading to potential exposure through drinking water.

- Consumer products: Cd can be present in some consumer products, such as jewellery, especially in items imported from countries with less stringent regulations.

Reducing Cd exposure is important to prevent health problems in the general population. To do this requires stricter regulations on industrial emissions, monitoring of food and water sources, and public awareness campaigns to discourage smoking and the use of Cd-containing products.

  1. What are the cytotoxic effects of Cd on liver and intestinal cells, and how do these effects relate to non-alcoholic fatty liver disease (NAFLD)?

Answer 2:

The aspects highlighted by the Reviewer in this point are of main importance for this study. Detailed information on Cd toxicity and pathogenic role in NAFLD/NASH are already reported in the introduction section (2nd par.). Below the Reviewer can find more details for his/her consultation:

Cd exposure can have cytotoxic effects on liver and intestinal cells, and these effects can be related to non-alcoholic fatty liver disease (NAFLD) in fact Cd can accumulate in the liver and directly damage hepatocytes, the primary cells of the liver. It can induce oxidative stress, disrupt cellular functions, and lead to cell death. Cd-induced liver cell damage can contribute to the development and progression of NAFLD. The oxidative stress and inflammation caused by Cd exposure can exacerbate the underlying metabolic factors that contribute to NAFLD, such as obesity and insulin resistance. Also, Cd can affect the gastrointestinal tract, particularly on enterocytes. It can disrupt the intestinal barrier function and lead to inflammation. The gut plays a crucial role in NAFLD development through the gut-liver axis. Disruption of intestinal cells by Cd can impair gut barrier function and increase the translocation of bacterial products (such as endotoxins) into the bloodstream. This can promote systemic inflammation and contribute to the progression of NAFLD. Moreover, Cd exposure can lead to chronic inflammation, both in the liver and systemically. Chronic inflammation is a key factor in the progression of NAFLD from simple fatty liver to more severe forms, such as non-alcoholic steatohepatitis (NASH). Furthermore, Cd exposure has been associated with insulin resistance, which is a significant factor in the development of NAFLD. Insulin resistance can lead to increased fat accumulation in the liver.

  1. How does the combination of Cd and fatty acids (FA) affect lipid droplet formation and ROS production in liver cells?

Answer 3:

This is another important point discussed in the premises to study hypothesis (2nd and 3rd par. of the introduction section), and then verified by the experimental data of this study.

For the Reviewer only, we add more information on this point: the combination of Cd and FAs can have complex effects on lipid droplet formation and ROS production in liver cells. FAs are ex-novo lipogenesis substrates; therefore, their supplementation to the liver cells induces triglyceride biosynthesis and storage in lipid droplets. When the liver cells are exposed to an excess of FAs, the accumulation of triglycerides within these droplets can induce a condition of cell damage named lipotoxicity; an increased production of ROS and the induction of stress pathways are characteristic hallmarks of this condition that may lead to an increased activation of cell death programs. Cd exposure can further enhance the lipotoxicity process by interfering with the lipid metabolism of the liver cell, namely with the activity of lipid metabolism enzymes and transcription factors that regulate cellular lipid accumulation and their toxicity effects. Therefore, the combination of Cd and FAs can have a synergistic effect on lipid accumulation and lipotoxicity phenotype of the liver cells. The lipid excess is the main cellular cue the development of non-alcoholic fatty liver disease (NAFLD). In addition, Cd exposure is known to induce the production of ROS in liver cells. Cd can disrupt the cellular antioxidant defence mechanisms, leading to oxidative stress. Also, FAs, especially when present in excess, can also contribute to oxidative stress by undergoing β-oxidation processes that generate ROS as byproducts. The combination of Cd and FA can lead to an additive effect on ROS production. Cd-induced oxidative stress can be amplified by the presence of FAs, and vice versa, leading to increased ROS levels in liver cells. Excessive ROS production can damage cellular components, including lipids, proteins, and DNA. In the context of the liver, oxidative stress is a significant contributor to inflammation and can further exacerbate NAFLD. The combination of Cd exposure and elevated levels of FAs in liver cells can create a negative environment characterized by enhanced lipid droplet formation and increased ROS production. This combination of effects can contribute to the progression of liver diseases like NAFLD and may be particularly harmful to individuals with underlying risk factors such as obesity, insulin resistance, or genetic predisposition.

  1. What role does melatonin (MLT) play in protecting cells from Cd toxicity, and how does it interact with Cd-induced oxidative stress?

Answer 4:

Melatonin’s role in Cd toxicity is introduced in the final sentence of the Introduction section and further information of this aspect is reported in the second part of the Discussion to support the findings of our study We believe that this part exhaustively explain the cytoprotective effects of this molecule in Cd toxicity, including its effects as antioxidant molecule.

We herein add other details for the Reviewer only; in our opinion these cannot be reported in the text being the length of the revised manuscript an issue to consider.

MLT plays a multifaceted role in protecting cells from cadmium toxicity. It acts as an antioxidant, reduces inflammation, preserves mitochondrial function, and may limit Cd uptake. By doing so, MLT helps counteract the adverse effects of Cd-induced oxidative stress on cells. Cd exposure can lead to increased ROS production in cells, and MLT can help counteract this by reducing the levels of harmful ROS. Cd exposure often disturbs activity of antioxidant enzymes, such as superoxide dismutase, catalase, and glutathione peroxidase, which are essential for maintaining the cellular antioxidant defence system and MLT can help restore their function. By reducing oxidative stress and limiting Cd's toxic effects, MLT can help promote the survival of cells exposed to Cd.

Moreover, Cd toxicity can impair mitochondrial function and lead to the generation of ROS within the mitochondria. MLT has been shown to protect mitochondria from Cd-induced damage, thereby preventing further ROS production within these organelles. In addition, Cd exposure can trigger an inflammatory response in cells and MLT has anti-inflammatory properties and can help modulate the inflammatory pathways, reducing the inflammation associated with Cd exposure. Also, Melatonin has been shown to have weak metal-chelating properties. While it may not chelate Cd as effectively as some other chelators, it can still reduce the uptake of Cd by cells to some extent, thereby limiting its toxicity.

  1. How do the results of this study contribute to our understanding of Cd toxicity and its impact on the gut-liver axis in the context of lipotoxicity and oxidative stress?

Answer 5:

We thank the Reviewer for rising this point that has suggested us to remove any direct indication to the gut-liver axis through the text. In fact, the liver and intestinal cells investigated in this in vitro study provide an experimental model that is too far from representing the lipotoxicity and oxidative stress effects of Cd and FFA on this axis.

Reviewer 2 Report

Comments and Suggestions for Authors

This manuscript showed toxicities of cadmium (Cd) combining fatty acids and/or melatonin in vitro. Some data may indicate toxicity of Cd and fatty acids. However, this manuscript does not show enough rationality and novelties.

(Major comments)

1. I cannot understand why melatonin was used to alleviate Cd- and/or fatty acid-toxicities. Melatonin is mainly synthesized in the brain, and I don’t know its physiological functions rather than regulation of the circadian rhythm. This means that melatonin may not be secreted to lessen oxidative stress; otherwise, secretion of melatonin to lessen oxidative stress may cause dysregulation of the circadian rhythms. Thus, the authors should explain the reason choosing melatonin and discuss association of the reason with the circadian rhythm. What do the authors try to address with the combination of Cd with melatonin? In the same manner, why were the intestinal epithelial cells assessed?

2. This manuscript does not indicate preserving mechanisms of melatonin. In the previous studies, some of the authors may investigate the apoptosis pathway and induction of antioxidant enzymes. At least, some of such indicators should be assessed. Furthermore, how much do the experiments in this manuscript reflect physiological events in vivo? Additionally, please indicate Cd concentrations in tissues of human body.

3. MAPKs seem to explain the mechanism. However, the explanation of MAPKs was difficult to understand and may not be conclusive. In Fig. 4, which MAPK was involved in the Cd toxicity and melatonin effect? The authors should conclude distribution of each MAPK in the cytotoxicity.

I cannot understand the intensities of phosphorylated protein bands should be divided by only those of the original protein bands: p-ERK 1/2, p-p38, or p-JNK should be divided by ERK, p-38 or JNK, respectively. Normalization by the intensity of GAPDH may not be required. Particularly, Cd seem to disturb energy homeostasis and possibly affect GAPDH (glycolysis protein).

4. In this manuscript, cellular viability, intracellular lipid accumulation, ROS generation, MAPK activities, and IL-6 secretion were measured. Coincidently, however, these indicators are possibly affected by Cd, fatty acids, and/or melatonin. In in vitro studies, to confirm influence of these indicators, signaling orders (pathways) of these indicators are often estimated using some inhibitors or agonists of these signaling molecules and/or other methods.

(Minor comments)

Line 64-65: This sentence is difficult to read.

Line 123: Please mention the vehicle of fatty acids. Was the vehicle added to the control medium in comparing to the cells treated with fatty acids?

Line 129: I cannot understand the meaning of “0.001% (v/v) H2O”.

Line 250-265 and 466-480: “SAPK” makes these sentence difficult to understand. (SPAKs in Line 250 may be SAPKs.) As far as Fig. 4, SPAK seem a synonym of JNK. Please define molecules of SPAK.

Line 303 and 453: Intracellular concentration of lipids is confirmed by synthesis, degradation, afflux, and efflux of lipids. In this study, however, lipid synthesis is not assessed, and this sentence may be invalid.

Line 343: I think that reduction of PA concentration is effective rather than protective effect of OA. Please describe the exact concentrations of “OA + PA”.

Line 448: This manuscript does not show any gut-liver axis (interaction), although the cellular responses to the cytotoxicity in Caco-2 cells partially accord those in the hepatocytes.

Fig. 2 and 3: Please confirm the control of the cell viability; the initial concentration or at the 0 micromolar of Cd?

Table 1: This table is difficult to understand. Was melatonin added to each condition except but Ctr and Ctr (DMSO)? Does “#p<0.05 versus CdCl2+MLT” mean “MTL 50 nM”? There are two levels of “CdCl2+OA+PA”.

Fig. 5(a): Please describe the units of the Y-axis. As far as I see, photos for the 50 micromolar-treatment show fewer cells than those in the less concentrations but do not show much red color, which may not suit Fig. 5(b).

Fig. 7: The reason why photos of 50 micromolar Cd + PA or +OA +PA were not shown should be explained. Additionally, lipotoxicity on Caco-2 cells should also be assessed.

Fig. 7(b): Do fatty acids induce ectopic fat accumulation in the intestinal epithelial cells? Please discuss these phenomena.

Fig. 8: Melatonin drastically diminishes ROS generation but slightly or does not alleviate cell viability. Does ROS generation indirectly reduce cellular viability?

That’s all.

Author Response

This manuscript showed toxicities of cadmium (Cd) combining fatty acids and/or melatonin in vitro. Some data may indicate toxicity of Cd and fatty acids. However, this manuscript does not show enough rationality and novelties.

(Major comments)

  1. I cannot understand why melatonin was used to alleviate Cd- and/or fatty acid-toxicities. Melatonin is mainly synthesized in the brain, and I don’t know its physiological functions rather than regulation of the circadian rhythm. This means that melatonin may not be secreted to lessen oxidative stress; otherwise, secretion of melatonin to lessen oxidative stress may cause dysregulation of the circadian rhythms. Thus, the authors should explain the reason choosing melatonin and discuss association of the reason with the circadian rhythm. What do the authors try to address with the combination of Cd with melatonin? In the same manner, why were the intestinal epithelial cells assessed?

Answer 1:

The reason why we investigated the effect of melatonin on Cd toxicity is that melatonin is a cytoprotective agent efficient in preventing Cd toxicity and liver damage in experimental models of fatty liver as well as in clinical trials on NAFLD patients. These aspects are better described in the revised manuscript, namely in final part of the introduction section and in the second part of the discussion section in which you can find the following sentence: “”.

We explained the reason why intestinal epithelial cells were investigated in this study in the introduction (end of the 2nd par.) and discussion section (end of 3rd paragraph) adding specific sentences as follows:

INTRODUCTION: “…as well as of other cellular elements that may contribute to harm the liver health [25,26], including intestinal epithelial cells that represent a natural barrier of the gastrointestinal tract and a system to preserve the liver homeostasis and function. To our knowledge, the effects of lipotoxicity on intestinal epithelial cells remains poorly characterized.”.

DICUSSION: “We also explored Cd and FFA toxicity in CACO2 intestinal epithelial cells since, to our knowledge, the effects of lipotoxicity on the gut epithelium remain poorly characterized although potentially relevant for the liver health [41].”.

  1. This manuscript does not indicate preserving mechanisms of melatonin. In the previous studies, some of the authors may investigate the apoptosis pathway and induction of antioxidant enzymes. At least, some of such indicators should be assessed. Furthermore, how much do the experiments in this manuscript reflect physiological events in vivo? Additionally, please indicate Cd concentrations in tissues of human body.

Answer 2:

We acknowledge the reviewer for this request of more data that are now available and add very important information for this study. These results are now presented in Figure 2, 7 and Suppl. S4.

Tissue-specific toxicity effects are described in the introduction section (end of the first paragraph), accompanied by relevant citations that describe specific mechanisms of toxicity including tissue concentrations, NOAELs, etc. Therefore, we prefer to avoid getting into more detail on these aspects in the text since these go beyond the scope of this in vitro study. However, below we report detailed information on Cd toxicity, tissue levels and concentrations utilized in in vitro studies for reviewer consultation.

Tissue Cd concentrations may vary on an individual bases and depend on the type of tissue and level of exposure. In human tissues (cadaveric study) these median levels ranged between 6.5 ug/kg of lens and 6800 ug/kg of kidney cortex, with liver containing 750 and the different intestine areas 100-110 ug/kg (determinations were in performed in wet tissues) Egger AE et al. Metallomics, 2019, 11, 2010-2019, DOI:10.1039/C9MT00178F. Another study on autopsies of non-poisoned Polish people the mean levels were: small intestine = 227, liver = 1540 and kidney = 16000 ug/kg (the latter correspond to 140 umol/kg) (Lech T & Sadlik J. Biol Trace Elem Res. 2017 Oct;179(2):172-177, doi: 10.1007/s12011-017-0959-5). A study on Australians without occupational exposure to metals showed liver and kidney cortex levels of 950, and 15450 ug/kg wet tissue and a corresponding mean urinary excretion of 2.30 ug Cd/g creatinine (Satarug S et al. Arch Environ Health. 2002 Jan-Feb;57(1):69-77, doi: 10.1080/00039890209602919).

A recent review article by SmereczaÅ„ski NM and Brzóska MM reported that the observed adverse effect levels (NOAELs) of Cd concentration in the blood and urine for clinically relevant kidney damage (glomerular dysfunction) are 0.18 μg/L and 0.27 μg/g creatinine, respectively, whereas the lowest observed adverse effect levels (LOAELs) are >0.18 μg/L and >0.27 μg/g creatinine, respectively, which are within the lower range of concentrations noted in inhabitants of industrialized countries (Int J Mol Sci. 2023 May 8;24(9):8413, doi: 10.3390/ijms24098413.). 0.18 ug/l correspond to 1.6 nM.

However, to induce significant toxicity in human liver cell lines, namely HepaRG and HepG2 cells (associated with reduced cell proliferation and increased ROS production), Cd concentrations have to reach micromolar concentrations (between 1 and 100 μM), whereas submicromolar concentrations (5-10 nM) promote liver cell proliferation (Niture S. et al. Environmetal Toxicol. 2023, https://doi.org/10.1002/tox.23731). In mouse liver cell lines significant reduction of cell viability is observed for concentrations > 2.5 uM with IC50 of approx. 20 uM (https://doi.org/10.1016/j.ecoenv.2022.114123). Studies in CACO-2 cells demonstrated that Cd concentrations <0.5 mg L−1 (4.4 uM) pose no toxic effects (Aziz R. et al. Biomed Res Int. 2014; 2014: 839538, doi: 10.1155/2014/839538). According with these studies, we confirm that in HepaRG and CACO-2 cell lines Cd toxicity is observed for concentrations > 20 uM (Suppl. Figure S1 and Figure 2).

Therefore, tissue and body fluid concentrations of Cd do not reflect those useful to recapitulate such toxicity in vitro in cell models.

  1. MAPKs seem to explain the mechanism. However, the explanation of MAPKs was difficult to understand and may not be conclusive. In Fig. 4, which MAPK was involved in the Cd toxicity and melatonin effect? The authors should conclude distribution of each MAPK in the cytotoxicity.

I cannot understand the intensities of phosphorylated protein bands should be divided by only those of the original protein bands: p-ERK 1/2, p-p38, or p-JNK should be divided by ERK, p-38 or JNK, respectively. Normalization by the intensity of GAPDH may not be required. Particularly, Cd seem to disturb energy homeostasis and possibly affect GAPDH (glycolysis protein).

Answer 3:

We are indebted with this Reviewer for the important observations on these data that stimulated us to perform more experiments and reorganize the presentation of the results in a more organized manner.

As suggested by this Reviewer, in the revised manuscript densitometry data phosphorylated protein bands were calculated dividing these for the original protein bands. Moreover, the new Figure 3 presents p-ERK ½, p-p38 and p-JNK expression data and including in supplementary material new results on the pharmacological inhibition of this MAPK which confirms the specific role of this kinase in MLT cytoprotective effect (Suppl. Figure S3).

  1. In this manuscript, cellular viability, intracellular lipid accumulation, ROS generation, MAPK activities, and IL-6 secretion were measured. Coincidently, however, these indicators are possibly affected by Cd, fatty acids, and/or melatonin. In in vitro studies, to confirm influence of these indicators, signaling orders (pathways) of these indicators are often estimated using some inhibitors or agonists of these signaling molecules and/or other methods.

Answer 4:

We fully agree with the Reviewer, in fact MLT was used to inhibit Cd and/or FFA toxicity considering its specific activity of MAPK modulator, and antioxidant and anti-inflammatory molecule. Again, new experiments with MAPK inhibitors are now included in the revised manuscript to ascertain the specific role of ERK-MAPK on Cd toxicity and MLT cytoprotective function (see answer n. 3).

(Minor comments)

Line 64-65: This sentence is difficult to read.

Answer: the sentence has been revised

Line 123: Please mention the vehicle of fatty acids. Was the vehicle added to the control medium in comparing to the cells treated with fatty acids?

Answer: This is now described in the section Materials and Methods, at the end of subsection “2.2.3 Cell treatments”.

Line 129: I cannot understand the meaning of “0.001% (v/v) H2O”.

Answer: This was since water was the vehicle of CdCl2.

Line 250-265 and 466-480: “SAPK” makes these sentence difficult to understand. (SPAKs in Line 250 may be SAPKs.) As far as Fig. 4, SPAK seem a synonym of JNK. Please define molecules of SPAK.

Answer: the acronym SAPK has been revised through the text and the definition is reported beginning of the 2nd par. of the Results.

Line 303 and 453: Intracellular concentration of lipids is confirmed by synthesis, degradation, afflux, and efflux of lipids. In this study, however, lipid synthesis is not assessed, and this sentence may be invalid.

Answer: we agree with the Reviewer and the sentence was revised substituting “lipid synthesis” with “neutral lipid accumulation”.

Line 343: I think that reduction of PA concentration is effective rather than protective effect of OA. Please describe the exact concentrations of “OA + PA”.

Answer: there was no reduction of PA final concentration during OA+PA treatment since the final concentrations of the two FFA remained the same as for the individual treatments, i.e. 200 µM each. This is specified in “Materials and Methods, section 2.2.3 Cell treatments”.

Line 448: This manuscript does not show any gut-liver axis (interaction), although the cellular responses to the cytotoxicity in Caco-2 cells partially accord those in the hepatocytes.

Answer: we revised this part accordingly (see Answer 1)

Fig. 2 and 3: Please confirm the control of the cell viability; the initial concentration or at the 0 micromolar of Cd?

Answer: this control test was cells not exposed to Cd (0 µM).

Table 1: This table is difficult to understand. Was melatonin added to each condition except but Ctr and Ctr (DMSO)? Does “#p<0.05 versus CdCl2+MLT” mean “MTL 50 nM”? There are two levels of “CdCl2+OA+PA”.

Answer: we converted this table in a bar chart plot and the reported combinations of treatments and compound concentrations are now described in detail.

Fig. 5(a): Please describe the units of the Y-axis. As far as I see, photos for the 50 micromolar-treatment show fewer cells than those in the less concentrations but do not show much red color, which may not suit Fig. 5(b).

Answer: the fact that few cells remain in the microscopy fields may give the impression that there is less red color at 50 µM Cd concentration, but after spectrophotometric analysis reported in 5(b) this red color (neutral lipid) accumulation is much more evident since it is normalized for the cell number.

Fig. 7: The reason why photos of 50 micromolar Cd + PA or +OA +PA were not shown should be explained. Additionally, lipotoxicity on Caco-2 cells should also be assessed.

Answer: in the previous version of the manuscript, we omitted these pictures in Figure 7 (now Figure 6) cause the cell layer was almost completely absent by the cell death effect of the combined treatment of Cd and PA. We included the pictures in the revised version of the Figure 6 (ex-Figure 7).

Fig. 7(b): Do fatty acids induce ectopic fat accumulation in the intestinal epithelial cells? Please discuss these phenomena.

Answer: ectopic fat accumulation may interest different tissues, including pancreas, muscle, etc., but to our knowledge the intestinal epithelium has not been studied in detail. Therefore, in this respect, we present original data. Furthermore, as reported in the Answer 1, the reason why intestinal epithelial cells were investigated in this study are now better described in the introduction (end of the 2nd par.) and discussion section (end of 3rd paragraph) adding specific sentences as follows:

INTRODUCTION: “…as well as of other cellular elements that may contribute to harm the liver health [25,26], including intestinal epithelial cells that represent a natural barrier of the gastrointestinal tract and a system to preserve the liver homeostasis and function. To our knowledge, the effects of lipotoxicity on intestinal epithelial cells remains poorly characterized.”.

DICUSSION: “We also explored Cd and FFA toxicity in CACO2 intestinal epithelial cells since, to our knowledge, the effects of lipotoxicity on the gut epithelium remain poorly characterized although potentially relevant for the liver health [Svegliati-Baroni et al FRBM 20129].”.

Fig. 8: Melatonin drastically diminishes ROS generation but slightly or does not alleviate cell viability. Does ROS generation indirectly reduce cellular viability?

Answer: Unfortunately, the cell viability test (MTT assay) was not sensitive enough to demonstrate the actual levels of cell death. Now we include apoptosis and necrosis levels assessed by FACS-scan analysis (Figure 7) that describe much better the effect of cell treatments including the response to MLT and its relationship with the ROS levels.

Reviewer 3 Report

Comments and Suggestions for Authors

Migni et al. in this work evidenced the Melatonin activity  as repairing agent in Cadmium- and free-fatty acid -2induced lipotoxicity.

The work is already published in Free Radical Biology and Medicine 201 (2023) 1–64  
https://doi.org/10.1016/j.freeradbiomed.2023.03.182 but mentioned

 some criticisms arise from the work:

1)       The concentration of Melatonin used in the experiments  was always fixed  at 50nM, please provide an explanation.  In biological condition the normal  range value  of Melatonin is equal to  10-60 pg/mL; therefore I  strongly suggest a new set of experiments using more realistic conditions to evaluate a potential concentration effect or a critical concentration as repairing agent of Melatonin.

2)       In cell lines, lipo-toxicity is further  induced by the palmitic and oleic acid treatments used at 200 micromolar for 48 hours.  Anyway the lipid accumulation has also been observed at lower concentrations and shorter times (i.e. Einaudi et al  Front Nutr. 2021 Nov 12;8:775382. doi: 10.3389/fnut.2021.775382).   Please provide an explanation about this point also in the discussion.

3)       ( line 78) the authors indicated “ cellular lipids”.. were measured, anyway they determined only their spectrophotometric dimension. The cellular  lipid characterization is missing, therefore the authors should have to add the fully  characterization if they refer to “cellular lipids” ( distinguishing the characterization  between the cytoplasmatic and membrane lipid classes and relative fatty acid content), otherwise the terms “ cellular lipids…” are not properly used. Therefore it’s better to move “ cellular lipids” from the beginning of line 78 to the end of sentence writing  “ in addition we determined  the cytoplasmic lipid accumulation  after the treatment only spectrophotometrically without performing   the  analytical characterization of  the cell  lipidome.

4)      The authors provided only the evaluation of  IL-6  as oxidative stress indicator; please explain the reason of this  choice and not of other inflammatory biomarkers;  the IL-6  is stimulated by prostaglandins ( ie PGE-2) so I suggest to include the prostaglandin measurement   performing  also  statistically correlation. Further  the ROS effect  on the “ cellular lipids”  are not reported .

 5)      Discussion:  please improve it

 6 ) Lines 444-445:  The experiments are performed in two different cell lines   
not “ in vivo”   (It’s reported  only the isolation of hepatocytes  
from the mice nothing related to the  intestinal cells ;
therefore this sentence is only a
assumption referred to this work.

Other comments:

 The reagents such as Melatonin, Palmitic Acid Oleic Acid are not mentioned  in the Section “ Materials”

 Concerning the Figures

Figure 1: Please increase the size of the characters

  In all figure several graphs are not totally readable. Please ameliorate them
( i.e  increasing the
line thickness and also the symbol size );
further please reorganize the figures not dividing the  graphs related to the same cell line
In the figure 2:please add the line related to  Melatonin trend  and perform the statistics also vs CTRLs  Figure 6:   concerning the graph “ E” please move  as a new Figure; in addition please
use different colors to evidence the lines afollowing also the previous indications  
Please  move  some graphs from main Text to SI. Please attention to the description of the captures that containing some errors Table 1: It’s more readable if the different kind  of the  treatments  are  reported  in vertical columns in order to easy compare the result of statics   Comments on the Quality of English Language

Some errors

Author Response

Migni et al. in this work evidenced the Melatonin activity as repairing agent in Cadmium- and free-fatty acid -2 induced lipotoxicity. The work is already published in Free Radical Biology and Medicine 201 (2023) 1–64 https://doi.org/10.1016/j.freeradbiomed.2023.03.182 but mentioned

Answer: This citation refers to an abstract of a poster presentation in which very preliminary data of this article were reported, essentially the set-up of the cell treatment protocol and a first series of data on Cd and FFA toxicity and MLT cytoprotection effect. We now include this citation in the revised manuscript (end of the Introduction section).

Some criticisms arise from the work:

1) The concentration of Melatonin used in the experiments was always fixed at 50nM, please provide an explanation. In biological condition the normal range value of Melatonin is equal to 10-60 pg/mL; therefore I strongly suggest a new set of experiments using more realistic conditions to evaluate a potential concentration effect or a critical concentration as repairing agent of Melatonin.

Answer 1: this concentration of MLT refers to pharmacological application of this hormanal substance (this type of utilization of MLT is now specified in the last paragraph of the Introduction section).

2) In cell lines, lipo-toxicity is further induced by the palmitic and oleic acid treatments used at 200 micromolar for 48 hours. Anyway the lipid accumulation has also been observed at lower concentrations and shorter times (i.e. Einaudi et al Front Nutr. 2021 Nov 12;8:775382. doi: 10.3389/fnut.2021.775382). Please provide an explanation about this point also in the discussion.

Answer 2: we revised the Discussion section accordingly (3rd par. from the top).

3) (line 78) the authors indicated “cellular lipids”.. were measured, anyway they determined only their spectrophotometric dimension. The cellular lipid characterization is missing, therefore the authors should have to add the fully characterization if they refer to “cellular lipids” ( distinguishing the characterization between the cytoplasmatic and membrane lipid classes and relative fatty acid content), otherwise the terms “ cellular lipids…” are not properly used. Therefore it’s better to move “ cellular lipids” from the beginning of line 78 to the end of sentence writing “ in addition we determined the cytoplasmic lipid accumulation after the treatment only spectrophotometrically without performing the analytical characterization of the cell lipidome.

Answer 3: we fully agree with the observation of this Reviewer and revised the sentence accordingly. The characterization of the cellular lipidome was beyond the scope of this study and cellular lipid accumulation was the actual aspect we investigated to mark the lipotoxicity process after Cd and FFA or MLT treatments.

4)The authors provided only the evaluation of IL-6 as oxidative stress indicator; please explain the reason of this choice and not of other inflammatory biomarkers; the IL-6 is stimulated by prostaglandins ( ie PGE-2) so I suggest to include the prostaglandin measurement performing also statistically correlation. Further he ROS effect on the “ cellular lipids” are not reported .

Answer 4: Unfortunately, we cannot assess prostaglandins, but we had the possibility to assess LO-5 protein expression which is directly involved in LTB4 metabolism. The results of this immunoblot study are now presented in Suppl. Figure S5. Other lipidomic studies, including ROS-induced lipid peroxidation, are beyond the aim of this study but surely will have all our consideration to plan for future studies. We acknowledge this Reviewer for pointing this out.

5) Discussion: please improve it

Answer 5: we agree with the Reviewer on the need to improve this section and more in general the entire manuscript; accordingly, we revised several parts of the manuscript and hope that this revised version will find the approval of this Reviewer.

6) Lines 444-445: The experiments are performed in two different cell lines not “ in vivo” (It’s reported only the isolation of hepatocytes from the mice nothing related to the intestinal cells ; therefore this sentence is only a assumption referred to this work.

Answer 6: the sentence has been revised accordingly.

Other comments:

7) The reagents such as Melatonin, Palmitic Acid Oleic Acid are not mentioned in the Section “ Materials”

Answer 7: a specific section has been included in the revised manuscript to describe these reagents.

8) Concerning the Figures

Figure 1: Please increase the size of the characters

In all figure several graphs are not totally readable. Please ameliorate them ( i.e increasing the line thickness and also the symbol size); further please reorganize the figures not dividing the graphs related to the same cell line. In the figure 2: please add the line related to Melatonin trend and perform the statistics also vs CTRLs. Figure 6: concerning the graph “ E” please move as a new Figure; in addition please use different colors to evidence the lines afollowing also the previous indications. Please move some graphs from main Text to SI.

Please attention to the description of the captures that containing some errors Table 1: It’s more readable if the different kind of the treatments are reported in vertical columns in order to easy compare the result of statics.

Answer 8: all the figures have been revised accordingly.

9) Comments on the Quality of English Language: Some errors

 Answer 9: The language and grammar have been revised.

Round 2

Reviewer 2 Report

Comments and Suggestions for Authors

This revised manuscript has been substantially improved according to the comments. However, I still have a concern that supraphysiological concentrations of Cd (and melatonin). As is often the case in in vitro study, supraphysiological stimulation is employed. Although it may be important to reveal mechanisms, it should be discussed with combination with in vivo studies; otherwise, at least it should be mentioned as a limitation of the study (please refer Line 509-510).

Additionally, I feel curious about some abbreviations that can be spelled out (for instance, should PTE be abbreviated?) or confusable because of overlapped abbreviation (please see the section 2.5).

Other points should be reconfirmed or modified as follows:

Line 21: The period should be deleted after the (50 nM).

Line 146: The comma after PD98 should be deleted.

Line 147-148 and 156: The suppliers of hydrocortisone is different.

Line 169: Please confirm the dimension of cell density (I think it /cm^2).

Line 235: A space should be inserted after Annexin V.

Line 257: “regent 2” may be “Reagent 2”.

Line 471-476: This sentence is quite complex, and should be separated to two or more sentences.

Line 480: What does “lipotoxicity induction effect” mean? Is it cytotoxic effect? It may be rational that FFAs show lipotoxicity which Cd synergizes with (Line 469-470 and other sentences). Does Cd have (but not enhance) lipotoxicity? In short, please define the “lipotoxicity”.

Only in the discussion section, tab space is inserted at the top of the paragraphs, which should be applied to the other section in the text.

That’s all.

Author Response

Query 1: This revised manuscript has been substantially improved according to the comments. However, I still have a concern that supraphysiological concentrations of Cd (and melatonin). As is often the case in in vitro study, supraphysiological stimulation is employed. Although it may be important to reveal mechanisms, it should be discussed with combination with in vivo studies; otherwise, at least it should be mentioned as a limitation of the study (please refer Line 509-510).

Answer 1: we completely agree with the Reviewer and in fact, during the first round of the revision process, we specified that the applied MLT concentrations refer to a pharmacological dosage (please refer to line 510-512), and in the last paragraph of the Discussion (line 515 onward) we included information on in vivo studies on MLT and Cd treatments that appear to confirm our mechanistic interpretation. However, we further revised this sentence according to Reviewer’s comment and suggested changes.

Other relevant information on Cd and MLT effects obtained in in vivo studies is reported in the first paragraph of the Discussion.

Query 2: Additionally, I feel curious about some abbreviations that can be spelled out (for instance, should PTE be abbreviated?) or confusable because of overlapped abbreviation (please see the section 2.5).

Answer 2: all the abbreviations have been checked as suggested.

Other points should be reconfirmed or modified as follows:

Query 3: Line 21: The period should be deleted after the (50 nM).

Answer 3: We thank the reviewer for the accuracy, we have corrected the error.

Query 4: Line 146: The comma after PD98 should be deleted.

Answer 4: We thank the reviewer for the accuracy, we have corrected the error.

Query 5: Line 147-148 and 156: The suppliers of hydrocortisone is different.

Answer 5: We thank the reviewer for the accuracy, we have corrected the error related to the company where we purchase the hydrocortisone.

Query 6: Line 169: Please confirm the dimension of cell density (I think it /cm^2).

Answer 6: We thank the reviewer for the suggestion. We have modified the text by including the cell density (cell/cm2) used for the MTT assay.

Query 7: Line 235: A space should be inserted after Annexin V.

Answer 7: In the updated version we have corrected the error.

Query 8: Line 257: “regent 2” may be “Reagent 2”.

Answer 8: In the updated version we have corrected the typo.

Query 9: Line 471-476: This sentence is quite complex, and should be separated to two or more sentences.

Answer 9: the sentence has been revised accordingly.

Query 10: Line 480: What does “lipotoxicity induction effect” mean? Is it cytotoxic effect? It may be rational that FFAs show lipotoxicity which Cd synergizes with (Line 469-470 and other sentences). Does Cd have (but not enhance) lipotoxicity? In short, please define the “lipotoxicity”.

Answer 10: "Lipotoxicity induction effect" refers to the process by which excessive levels of lipids in cells or tissues lead to cellular damage or dysfunction. When there's an overload of certain types of cellular fats, particularly saturated fatty acids, this can disrupt normal cellular processes, leading to toxicity.

This overload of lipids can interfere with cellular functions like insulin signaling, mitochondrial function, and various metabolic pathways. It often results in the production of ROS and inflammation, which can ultimately cause damage to cell membranes, organelles, and even cell death. Lipotoxicity is often associated with conditions like obesity, diabetes, and cardiovascular diseases where there's an abnormal accumulation of lipids in tissues like the liver, heart, or pancreas. Managing lipid levels through diet, exercise, and medication can help mitigate the effects of lipotoxicity.

In the revised manuscript, we described in detail the lipotoxicity hallmarks considered in our study (from line 486 onward).

Query 11: Only in the discussion section, tab space is inserted at the top of the paragraphs, which should be applied to the other section in the text.

Answer 11: We thank the Reviewer for the advice. In the updated version we have included tab space throughout the text in the various sections.

Reviewer 3 Report

Comments and Suggestions for Authors

No additional comments.

Comments on the Quality of English Language

N/A

Author Response

We thank the reviewer for accepting our changes to the paper as suggested by him and the other reviewers.